# Protestant Congregational Song in the Philippines: Localization through Translation and Hybridization

**Glenn Stallsmith**

The Divinity School, Duke University, Durham, NC 27708, USA; glenn.stallsmith@duke.edu

**Abstract:** Historically, the language of Protestant congregational song in the Philippines was English, which was tied to that nation's twentieth-century colonial history with the United States. The development of Filipino songs since the 1970s is linked to this legacy, but church musicians have found ways to localize their congregational singing through processes of translation and hybridization. Because translation of hymn texts from English has proven difficult for linguistic reasons, *Papuri*, a music group that produces original Tagalog-language worship music, bypasses these difficulties while relying heavily on American pop music styles. Word for the World is a Pentecostal congregation that embraces English-language songs as a part of their theology of presence, obviating the need for translation by singing in the original language. Day by Day Ministries, the third case study, is a congregation that translates beyond language texts, preparing indigenous Filipino cultural expressions for urban audiences by composing hybridized songs that merge pre-Hispanic and contemporary forms.

**Keywords:** translation; worship music; pentecostalism; Tagalog language; colonialism; hybridization; Protestant mission

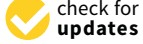



## 1. Introduction: Missionization and Language Use in the Philippines

The history of Christianity in the Philippines is a story of colonialism. The country's relationships with two primary colonial powers may be abbreviated into a statement such as: "Three hundred years under Spain and fifty years under America". Philippine Christianity is inextricably tied to forms of the faith that were brought from these two governing powers, with Spain controlling the nation from the sixteenth to nineteenth centuries and the United States from the early- to mid-twentieth century. Roman Catholicism was introduced by the first wave of Spanish colonizers and has since been the nation's primary religious expression. The defeat of Spanish forces by the United States in the Western Hemisphere led to the 1898 Treaty of Paris, in which the Philippines were awarded to an American nation that was just beginning its own venture into international imperialism. Squelching a nascent independence movement, U.S. forces in the Philippines subdued a series of nationalist uprisings and made indelible marks in the educational system and in the religious landscape, sending both American teachers and American missionaries at roughly the same time.[1] Although many American Protestant and evangelical groups have exerted significant influence in the years since, the Philippines remains majority Roman Catholic, with estimates ranging between 70 and 80 percent of the national population.[2] This article focuses on the 10 to 20 percent of Filipinos who consider themselves Protestants or evangelicals, for which ties to American churches remain strong through systems of funding and cultural influence.

These two respective colonial powers had differing approaches to the indigenous languages of the Philippines. Unlike with their colonies in the Americas, Spain chose not to teach Spanish widely as a lingua franca in the Philippines. It remained a language of the elite, with indigenous languages vying for predominance in the church and the marketplace. Tagalog, the language of the people who lived near what is now the capital

city of Manila, rose in prominence due to its geographic location near the centers of colonial power. Even now, the decision to make Tagalog the national language still sparks controversy, with speakers of other regional languages, such as Ilocano and Cebuano, contesting its primacy.[3] The American takeover of the Philippines did not replace the use of indigenous Philippine languages in everyday life, but English became the language of education and governance, giving it an outsized role relative to the comparatively short time that it has had to take root.[4]

Though English has taken over as the language of the legislature, the courts, and the school systems, most Filipinos continue to use indigenous languages in other life settings, including the church. While it is impossible to deny the role that Spanish and English have played in the life of Philippine congregations, the languages of Christian worship have always included the nation's over one hundred indigenous tongues. This is certainly true of the lyrics used in church music. As musicologists David Kendall and David Irving have demonstrated, liturgical songs developed by Catholic clergy over three hundred years included a mixture of indigenous and Spanish elements that combine musical and linguistic features of colonizer and colonized.[5] These hymns and service music pieces were set to texts in a variety of languages—Latin, Spanish, and several Philippine languages—and the forms of the songs, while dominated by European instruments and subsequent methods of creating scores, nonetheless contained elements of indigenous music-making. The work of these scholars shows that, despite the major cultural and linguistic influences exercised by Spain and the Roman Catholic Church, the composition and performance of church music in the Philippines has always gone through various processes of localization.[6]

The focus of this article is more recent and, in many senses, narrower. By looking at the processes of localization in the praise and worship movement over the past fifty years, my scope is thereby limited to Protestant and evangelical churches that have been largely influenced by American missionization and English language songs. After World War II, the Philippines became an independent nation, but cultural ties to the United States remained strong, especially in the worship music used in Protestant and evangelical churches. The three cases in this article each examine a different emphasis of the localization that took place during this post-war period, beginning with the problems of translating early-twentieth-century English-language hymns into Tagalog. The first case shows how the music project known as *Papuri* created original songs in the national language starting in the 1970s to bypass these translation pitfalls, thereby creating a localized genre of worship music that combines natural-sounding lyrics and Western pop music idioms. The second case focuses on the Pentecostal movement in the Philippines, which in the 1980s embraced English-language songs, obviating the need for translation altogether. Looking at the Word for the World congregation, we see how some Christians moved past the language of the lyrics and concentrated instead on a specific hermeneutic of reading Old Testament worship texts. The third case investigates a Filipino church, Day by Day Ministries, that since the 1990s has created hybrid praise and worship compositions by localizing indigenous Philippine music and dance forms. When texts of these songs are translated, they tend to move from indigenous languages to Tagalog, not from English to Tagalog. Day by Day has pioneered a unique method of localizing indigenous music and dance forms for urban audiences, for the purpose of imagining a non-Hispanicized version of Philippine Christianity.

## 2. *Papuri*: A Local Solution to the Problems Posed by Translation

The arrival of American missionaries to the Philippines in the early twentieth century added English to an already-complex language situation. Similar to the teachers who assumed that education must start with the English language as both a subject and a means of instruction, American missionaries introduced hymns that they did not attempt to adapt musically. There is little evidence that serious attempts were made by these missionaries to encourage the creation of indigenous worship music (for some, it was even heretical to suggest that "pagan" music could be used for the purpose of praising God). The

primary mode for creating worship songs for Protestant churches of the Philippines was translation—or, more accurately, adaptation.[7] Almost all attempts to localize congregational songs during most of the twentieth century assumed that the tunes would remain the same, meaning that changes needed to come in the lyrical content.

An example of one such adaptation is the hymn "Great Is Thy Faithfulness", composed in English in 1923 by Thomas Chisholm and William Runyan. The most popular Tagalog adaption of this hymn is titled "Tunay Kang Matapat".[8] The first stanza and refrain reveal that Filipino translator Max Atienza followed the general meaning of Chisholm's original words:

| Original, by Chisholm | Tagalog version, by Atienza | My translation of the Tagalog |
|---|---|---|
| Verse 1: | | |
| Great is thy faithfulness, | Tunay kang matapat, | Truly you are faithful, |
| O God my Father. | Dios nami't Ama, | God our Father. |
| There is no shadow of turning with thee. | Di nagbabago di nagiiba; | (You) are not changing or becoming different. |
| Thou changest not, thy compassions, they fail not. | Mahapon, matanghali, maumaga, Sa buong panahon, | Afternoon, noon, morning, All the time you are faithful. |
| As thou hast been thou forever wilt be. | matapat Ka. | |
| | | |
| Refrain: | | Truly you are faithful, |
| Great is thy faithfulness! | | Truly you are faithful, |
| Great is thy faithfulness! | Tunay kang matapat, | Every day I see. |
| Morning by morning new mercies I see. | Tunay kang matapat, Araw araw aking namamalas. | |
| All I have needed thy hand hath provided. | Ang iyong kabutihang walang kupas; | Your goodness never fades. Truly you are faithful, in (your) protection. |
| Great is thy faithfulness, Lord, unto me! | Tunay kang matapat, sa paglingap. | |

Since the melodic lines could not be changed, the person adapting the text into Tagalog did not have the freedom to add or subtract notes to create space for different words. Not only does this make word-for-word translation nearly impossible, it also makes creative adaptations such as "Tunay Kang Matapat" difficult to indigenize. Words are made up of syllables, which in turn are built of consonant and vowel sounds. A spoken word, however, makes meaning when those disparate sounds become more than the sum of their respective parts. This requires that the syllables exist in a stable relationship with each other, so that the pronunciation of a given word stresses specific syllables in relation to the others. This stress can happen by lengthening the amount of time a given syllable is sounded relative to the syllables around it. Stress can also be accomplished in the form of loudness, with one syllable in a word being spoken more strongly than others. In real-life speech patterns, stress is usually heard as some combination of loudness and length. An example from English is the difference between the words "missionary" and "machinery". Each of the four syllables in these two words is nearly the same when considered in isolation, but when they are pronounced together as one word, it is the stress given to a specific syllable that distinguishes one from another. For "missionary", the stress is placed on the first syllable; in the case of "machinery", the stress is on the second syllable.

Because the meanings of words in Tagalog and other Philippine languages are dependent on placing the correct stress on a given syllable, this becomes important in the process of adapting song lyrics to an existing hymn tune. In the case of the word *matapat*, the everyday spoken word receives its stress on the ultimate syllable: *ma-ta-PAT*. When setting the Atienza text to the 1923 tune by Runyan, however, there is a stress placed on the first syllable when *ma-* lands on a dotted quarter note in the first line of the hymn. To

a speaker of Tagalog, this makes the word sound like it is being mispronounced. While most Christian worshipers can rationalize this awkward construction by considering the exigencies of hymn text adaptation, it nonetheless sounds like what it is: a foreign tune that has had an adapted local text mapped onto its notes.

This awkwardness was not enough in itself to move Protestants in the Philippines to start a sustained movement of creating new worship songs of Tagalog texts set to original tunes. It would take a decree in 1977 by the national broadcasting regulator—Kapisanan ng mga Brodkaster sa Pilipinas (KBP)—that all Philippine radio stations must air at least one original Filipino composition every hour. In response, the Far East Broadcasting Company (FEBC), a nation-wide network of Protestant radio stations, launched a series of song-writing competitions, with the best submissions broadcast on air. It was soon discovered that these compositions could be compiled and sold on cassette tapes, thus launching the ministry of *Papuri*, which became a record label synonymous with this growing corpus of songs. The musicians who performed on the recordings also formed acts that toured the country in series of live concerts.

*Papuri* still actively creates and distributes new Tagalog language songs, but the height of its popularity came during its first decade of existence. These new songs were a novelty, finding their way into churches and complementing the existing corpus of Western songs. They became fixtures as congregational songs, pieces for choirs, and "special numbers" for small ensembles and soloists. One of the most beloved, even to this day, is "Sapagkat ang Diyos ay Pag-ibig" (Because God Is Love), from the project's first album in 1978:

| Original Tagalog Lyrics | My translation |
| --- | --- |
| Verse 1:<br>Pag-ibig ang siyang pumukaw<br>Sa ating puso at kaluluwa<br>At siyang nagdulot sa ating buhay<br>Liwanag sa dilim at pag-asa | Love is what inspires<br>our hearts and souls<br>and is what gives purpose to our lives<br>light in the darkness and hope. |
| Verse 2:<br>Pag-ibig ang siyang buklod natin<br>'Di mapapawi kailan pa man<br>Sa puso't diwa, tayo'y isa lamang<br>Kahit na tayo'y magkawalay | Love is what binds us<br>without fading.<br>In heart and spirit, we are one<br>even when we are apart. |
| Chorus:<br>'Pagka't ang Diyos natin, Diyos ng pag-ibig<br>Magmahalan tayo't magtulungan<br>At kung tayo'y bigo, ay 'wag limutin<br>Na may Diyos tayong nagmamahal | Because his is our God, God of love,<br>Let's love one another and work together.<br>And if we fail, don't forget<br>That we have a loving God. |

"*Sapagkat ang Diyos ay Pag-ibig*", originally composed in Tagalog and set to its own melody, is free of any awkward pairings between tune and adapted text. However, though the lyrics are sung in a natural way, with the correct syllables stressed, this song shows ways in which the *Papuri* project focused its localization of Christian songs on the lyrics. The original studio recording of this song, from the album *Papuri #1*, is solidly in the idiom of Western popular music of the 1970s.[9] The song begins with an introduction of strings and piano before a group of men sing the first two lines of the verse in unison. On the third line, women's voices join those of the men. The chorus combines men's and women's voices, with some lines breaking into different harmonies, accompanied by the piano. A viola and a violin play the melody during a two-line bridge, supported by runs on the piano. None of these musical components are outside of standard Western popular music of the period, including a standard major-key chord progression. "*Sapagkat ang Diyos ay Pag-ibig*" shows that, despite the attempt to improve on the translation and adaption work

of previous generations, the work of localization in this era mostly did not extend to the musical sounds of the compositions.[10]

### 3. Word for the World: Transcending the Local through the Spirit

While songs by *Papuri* were being developed and adopted by churches throughout the Philippines, a parallel movement was taking place. A new set of songs known as "praise and worship" was coming from the United States, and the churches that adopted these songs had a new set of expectations about the purpose of worship music. Rather than seeing the message of their songs as linguistically coded, the Pentecostals behind this worship movement believed that God was revealing a new biblical theology of worship itself. Whereas the standard Protestant approach to worship music had been to illuminate an understanding of the scriptures, this new movement of praise and worship saw the songs themselves as instrumental in drawing the worshiper to God, without linguistic mediation. Translation of words into a new language was an important tactic for early generations of missionaries, but the practitioners of this emerging praise and worship style did not prioritize translation in the same way. The case in focus here is a Church of God congregation called Word for the World Christian Fellowship, which was begun in the early 1980s by American missionaries in Makati City, a wealthy section of the Manila capital region.[11] To understand why this Pentecostal church embraced and adopted English-language songs more fully than songs in Tagalog, we need to look at the North American origins of this style of worship.

In the late 1940s, Reg Layzell, a Canadian Pentecostal pastor, began to teach his parishioners that the scriptures contained concrete instructions by which Christians could experience God's presence. In what became known as the Latter Rain revival, Layzell and his followers interpreted specific verses from the Old Testament as divine commandments about how one should approach the heavenly Father. Key verses that fed this sacramental perspective of praise and worship music were Psalm 22:3: "But thou art holy, O thou that inhabitest the praises of Israel" (KJV), and Psalm 100:4: "Enter into his gates with thanksgiving, and into his courts with praise: be thankful unto him, and bless his name" (KJV). Layzell's reading of these verses was simple: if a Christian praises God, then God will be present (Lim and Ruth 2017, p. 112).

Later, in the 1960s, as the revival spread across Canada and the United States, readings of additional Old Testament texts led these Latter Rain Pentecostals to prescribe musical practices that would manifest God's presence. Scripture passages about various structures for worship—Moses's tabernacle, David's tabernacle, and Solomon's temple—were read as models for how God's people should worship in the twentieth century.[12] The fundamental idea behind these typologies involved a progression from praise to worship, with the worshipers imagining themselves as beginning outside the structure—either the tabernacle or temple—and moving through the interior spaces before arriving at the ultimate place of God's presence: the Holy of Holies. This imagined journey always followed the same pattern: entering the building with praise (see Psalm 100:4), moving to a time of thanksgiving and adoration, and finally finishing in an attitude and posture of "worship".[13] The extent to which this progression moved more quickly or slowly depended on several circumstances, but the achievement of the final goal was never in doubt: if God's people praised and worshiped God, then God's presence would be manifest among them. This Latter Rain teaching took root among Pentecostal networks, both formal and informal, and spread throughout the world, accompanied by the genre of music known as "praise and worship".[14] The label of this style of music was meant to show the biblical theology that undergirded it; if Christians would first praise, then God's presence would be manifested in their gathering, thus allowing true worship to take place. This progression began to drive the choice of songs used in worship services, with faster, more upbeat songs starting the service in the praise mode before moving to slower and more contemplative musical settings.[15]

This teaching about praise and worship came to the Philippines from the United States in the 1980s. Musicians and producers from Integrity's Hosanna! music label, an organization with ties to the Latter Rain movement, visited Manila to conduct workshops about their biblical theology of worship. These included recording artists such as Lamar Boschman, Ron Kenoly, and Don Moen, and their seminars coincided with extensive growth in Pentecostal and charismatic churches in the Philippines during the 1980s. The architecture-based progression from praise to worship became predominant, first among Pentecostal churches such as Word for the World, and then spreading throughout a variety of Protestant denominations.[16] Church musicians were likened to the Israelite tribe of the Levites who attended to worship practices as prescribed in the Old Testament.[17] These Filipino musicians learned to categorize songs according to their place in the progression of the worship service—"thanksgiving" songs always opened the service, and "throne room" songs always ended the time of worship that preceded the preaching. Some songs were even labeled according to specific furniture placed between the entrance and the inner courts, with "prayer songs", for instance, set aside for the moments spent at the altar of incense (cf. Exodus 30) on the way to the Holy of Holies.

Unlike the other two cases presented in this article, Word for the World is less a hub of innovation in worship music in the Philippines than an exemplar of praise and worship's influence. It demonstrates, for instance, how a Pentecostal theology of worship led to a different perspective on the translation of song texts from *Papuri*. At Word for the World the primary congregational songs were unadapted and untranslated English hymns and choruses, but this was not because Tagalog-language songs were considered unimportant.[18] It was not the "translation" on the lexical level that Filipino Pentecostals were mostly concerned with; rather, they were focused on the communication of theological principles to a direct and personal encounter with God. There are certainly sociolinguistic reasons for the privileging of English songs, e.g., the role of American missionaries in launching the church or the desire to attract worshipers who were comfortable speaking English at work and school, but it would be a mistake to see the use of untranslated songs as merely the preference of the congregation's American founders (Appadurai 1996, p. 90). Word for the World was among a movement of churches trying to bring people into a direct encounter with God, and this involved a complex set of practices that largely transcended cultural forms like the use of specific languages or dialects. In short, songs of "praise" and songs of "worship" had their own potency, if programmed correctly, to bring people into God's presence. This power was primary, and it was more important than the worshipers' ability to comprehend the lyrics of songs.

This subordination of linguistic forms to biblical typology can be understood if one views Pentecostalism itself as a "hard" cultural form. Arjun Appadurai, who coined this formulation, compared hard cultural forms with others he considered "soft". The former "come with a set of links between value, meaning, and embodied practice that are difficult to break and hard to transform".[19] Soft cultural forms, by contrast, have looser links between embodiment and meaning; they are more easily translated or adapted. Ethnographers and anthropologists often describe Pentecostal worship services as following nearly identical forms in the various places they are found throughout the world.[20] The theological reasons for this "hardness" are found in Pentecostalism's ways of reading the scriptures. As noted above, many Pentecostals read the Old Testament's descriptions of an ancient tabernacle structure as containing the timeless instructions for worshiping that transcend any contemporary liturgical forms. This is consistent with a reading of the New Testament book of Acts, which views the work of the Holy Spirit as countercultural and transcultural. For instance, the spiritual gift of speaking in tongues, or *glossolalia*, is practiced as the utterance of unknown—and some would say, non-human—languages in the course an ecstatic encounter with the Spirit. The point of such an outburst is not to speak the language of the hearer; rather, it is to transcend normal communication mechanisms altogether and demonstrate a direct working of divine power. In this way, the ecstatic form of speaking in tongues will sound the same in Manila as in Los Angeles, with the speaker's words as

inscrutable in one place as in the other. As Paul Freston puts it, Pentecostalism's forms are beyond translation, becoming a "culture 'against culture.'" (Freston 2013, p. 66).

In this way, the worshipers at Word for the World who sing English songs are being entirely consistent with Pentecostalism's commitment to a noncontextualized view of worship practices. This idea of otherworldly transcendence is found in one of the popular praise and worship songs that the congregation sang in the 1980s: "Be Exalted, O God", composed by New Zealander Brent Chambers:

> I will give thanks to Thee
>
> O Lord, among the people.
>
> I will sing praises to Thee
>
> among the nations.
>
> For Thy steadfast love is great,
>
> is great to the heavens
>
> and Thy faithfulness,
>
> thy faithfulness to the clouds.
>
> Be exalted, O God
>
> above the heavens.
>
> Let Thy glory be over all the earth.

The first-person perspective in the song's lyrics indicates that the singer will sing "among the people" and "among the nations". However, the focus is not on speaking *to* the people themselves in the various languages of the nations. Instead, the worshipers are speaking to God, who is described as "above the heavens" and whose glory is "over all the earth". This is very different from the contextual church growth model of missionization that flourished in the US during the middle of the twentieth century. Purveyors of that strategy, the most prominent being Donald McGavran, asserted that the primary barriers to Christian conversion were problems of communication: missionaries who did not understand the language of the people; worship services that contained inappropriate forms of gathering; and congregations that attempted to assemble diverse and, therefore, antagonistic sets of culture groupings in the same service (McGavran 1990). Churches influenced by strategies based on contextualization saw language translation, especially of the scriptures, as of utmost importance in the task of evangelization. Pentecostalism, by contrast, had a different perspective on the relative importance of language itself. The use of the English language with songs in the Philippines, therefore, was not incongruous with the set of practices that accompanied the direct progression of the worshiper from the outer courts to the innermost Holy of Holies. Language use was not always localized in Filipino Pentecostal churches because the Holy Spirit was thought to work apart from one's perception of linguistic codes.

## 4. Day by Day Ministries: Becoming Local through Hybridization

For yet another approach regarding translation and worship music in the Philippines, we turn to Day by Day Ministries, a congregation launched in the 1980s. This church was initially forged in an international context, originally made up of Filipino expatriate workers in Saudi Arabia. The founding pastor, Ed Lapiz, was working at a hospital in that country when he began a Wednesday night Bible Study for other Filipinos. This weekly gathering grew into a worshiping congregation, and given the transient nature of overseas work, Lapiz desired to establish a base in the Philippines which could reach the Wednesday night participants once they repatriated to their home country. Day by Day was thus launched in the Philippines in the late 1980s, with Lapiz returning from Saudi Arabia in 1989 to lead the growing church.

Perhaps sparked by the experience of living as foreigners in another nation, Day by Day has sought to worship through inculturated forms that represent an essentialized

version of Philippine Christianity. Early in the 1990s, Day by Day began to develop unique worship songs that went beyond the Tagalog text adaptations of the early Protestants or the original compositions such as those by *Papuri*. They instead sought to create new songs based on a uniquely Filipino style that borrowed from indigenous sounds, movements, and material culture. Much of this innovative work was spearheaded by a troupe of musicians and dancers called KALOOB, which formed in 1994. The task of this organization, which works under the umbrella of Day by Day, is to research traditional music and dance of the Philippines so that it can be represented and, in a sense, translated for congregations and audiences throughout the nation. KALOOB does this through two primary modes: (1) "prayformances" that recreate indigenous practices for concert audiences and (2) new hybridized songs that combine indigenous components of rhythmic elements, melodic motifs, and instrumental ensembles with praise and worship structures.

Both of these modes of creation begin with extensive periods of research among indigenous Filipino communities. Lapiz, who obtained an M.A. and Ph.D. in Philippine Studies from the University of the Philippines, has trained KALOOB participants in ethnographic research methods.[21] Group members interview culture bearers from various ethnolinguistic groups to learn traditional forms of singing and dancing.[22] Then, in the case of "prayformances", KALOOB seeks to accurately represent these forms in live performances so that other Filipinos may observe and experience folkways that predate the colonizing influences of Spain and the United States. The prayformances are not liturgical, per se, but they are offered to the public in a "spirit of prayer" as a way of connecting contemporary Filipinos with modes of music and dance that are indigenous to these islands.[23] In this sense, the work of translation moves in multiple directions: from the rural to the urban and from the indigenous peoples of the highlands to the lowlanders of the coastlands. The songs and dances presented for these audiences are modified for the stage, but the intent is that the essential forms are performed as closely as possible to those learned from the culture bearers themselves.

The second mode of KALOOB's adaption work comes in the form of newly composed worship songs that draw on these same researched cultural materials. In contrast to the performances designed for audiences, these congregational songs are meant to be sung during worship services at Day by Day. They are, therefore, hybridized in ways that allow singable melodies to accompany musical components from indigenous societies. An example of this style of original composition is "U Javih—Apo Dios".[24] The studio-produced recording of the song opens with the interlocking sounds of handheld gongs that are played at traditional celebrations among several indigenous societies in the Cordillera mountains of the northern Philippines. This is followed by a group of women singing, in unison, a pentatonic melody from that region, one of several that make up a genre known as *salidummay*. Their tune is supported by the ongoing pattern of the gongs, to which a drum has been added. The lyrics the women are in one of the Kalinga languages that is indigenous to that region:

| U Javih, Apo mi | O Yahweh, our God |
| Napatog nan ngajen nu | Your name is great |
| Apo inganat-ingana | God forever |
| Lagsakan ji napatug un Apo | The mighty God is praised[25] |

After two repetitions of this introductory verse, the flourish of a traditional bamboo flute is heard in the mix. Then a group of men singers enter in unison, supported by the same ongoing rhythmic pattern, this time singing in Tagalog. At this point synthesized instruments are added, supporting what sounds like a shift to a Westernized sound. This change allows the rest of the song to be sung by the entire congregation, and the song takes on a familiar structure of verse, pre-chorus, and chorus. In this case, each of these three sections becomes progressively more singable, with the chorus being the most accessible for lay singers to join:

| *Verse 1:* | |
|---|---|
| Kung aming sasariwain | We are renewed |
| Pagdaloy ng lumipas na kasaysayan | in the flow of passing history |
| Mula pa nang nililok n'yo | ever since you created |
| Ng labi ang larawan ng sandaigdigan | from your lips the form of the universe. |
| | |
| *Pre-chorus 1:* | |
| Sadyang nanatili | Your nature |
| Sadyang 'di nagbago | always remains (and) |
| Ang kalikasan n'yo | never changes. |
| O Apo Dios | O Lord God[26] |
| Sadyang 'di mawari | The mystery of your greatness |
| Sadyang 'di mantanto | is steadfast but not perceived |
| Hiwaga ng kadakilaan n'yo | nor grasped. |
| | |
| *Chorus:* | |
| O Apo Dios (6x) | O Lord God |

This pattern repeats again, with different lyrics for a second verse and second pre-chorus. As the repetition of the chorus fades out for the final time, the original ensemble of women vocalists is heard again, singing the Kalinga-language text accompanied by the gong pattern that opened the recording. This creates an overall A-B-A form, with even the most musically untrained listeners able to discern clear differences between the A and B sections in terms of language, melodic structure, rhythm, and instrumental timbre.

I argue that this recording of "U Javih—Apo Dios", with its Kalinga text and indigenous musical sounds bracketing a standard praise and worship structure, is itself an act of translation. The words of the lyrics themselves are not translated, per se. That is, the praises of God that are sung in Kalinga at the beginning and end, while echoed in the Tagalog section, are not strictly translated. The work of translation instead happens on a different level, in which indigenous cultural artifacts, in the form of specific Kalinga instruments and sonic structures, are presented to lowlander and urban audiences in a performance of essential Filipino-ness. Day by Day's congregations in Manila and other cities in the Philippines are offered this song as a mechanism for participating in a worship form that predates colonial influences from the Western Hemisphere. Without explicitly stating it in the lyrics, the overall sound of the song itself proclaims that Filipinos can worship God as Filipinos, and this includes the aspects of cultural heritage that preceded the arrival of the so-called Christianized songs and prayers of the multiple waves of foreign missionaries. This claim, more than the Kalinga and Tagalog lyrics, is what is being "translated" for the congregation.

The performances (and prayformances) of KALOOB are local attempts to subvert the hybridized church songs that have been the centuries-old products of colonialization. Songs such as "U Javih—Apo Dios", with their mixture of highland and lowland musical elements, are designed to allow people from across the Philippines to experience a decolonialized way of worshiping. For much of the lowland Philippines, indigenous cultures were merged with those of the colonial powers in a process that musicologists have called "Hispanicization".[27] This hybridization, however, often felt like a complete replacement of the original music styles, and nationalist movements in the Philippines since the nineteenth century have tapped into highland and Muslim indigenous music forms as a way of reestablishing an essential Filipino sound.[28] The worship songs of Day by Day and KALOOB can be seen as continued attempts to create hybridized forms that assert a precolonial Filipino essence, but these worship songs are composed with the congregation in mind. They are an accessible way for evangelical Christians to see themselves as part of an ancient culture. The translation of texts, while present in some pieces of Day by Day's corpus, is subordinated to a re-presentation of traditional music forms.

This corpus of hybridized songs is generally acknowledged as a worthwhile innovation by Filipino Protestant Christians, with some of Day by Day's songs being sung in other congregations. Not all Filipinos are as concerned about cultural renewal or revitalization as are Lapiz, KALOOB, and Day by Day, but most recognize these compositions as at least an interesting curiosity. The main challenge facing the widespread adoption of this corpus is, of course, its hybrid nature. Indigenous Filipino Christians, in a sense, do not need these reimagined versions of their own traditional music practices. Urban Filipinos, on the other hand, are often satisfied by the Tagalog and English songs available in their churches and across multiple media outlets. For Pentecostals especially, the desire is for worship expressions that transcend local, or any, cultural expressions.

## 5. Conclusions

These three cases provide a fuller understanding of the scope of translation in congregational song in Christian worship. In the case of the songs composed by *Papuri*, the exigencies of fitting multisyllabic Filipino words into a given melody show that translation from European languages is nearly impossible. The new *Papuri* compositions that responds to this difficulty, even if the music mimics a Western idiom, provide an opportunity to communicate more clearly than the mispronounced words of translated hymns. For Pentecostal Filipinos who worship in congregations such as Word for the World, the object of translation is not the content of words but rather the Spirit-guided forms of worship that transcend specific cultural expressions. Nor is the translation of specific texts the focus of the hybridizations performed by worshipers at Day by Day Ministries. Unlike for most Pentecostals, local cultural forms are of great importance in this congregation, and the essence of indigenous songs and dances are portrayed in music and movement, with the work of translation coming in the reframing and re-presentation of precolonial artistic forms.

**Funding:** This research received no external funding.

**Institutional Review Board Statement:** Not applicable.

**Informed Consent Statement:** Informed consent was obtained from all subjects involved in the study.

**Data Availability Statement:** Not applicable.

**Conflicts of Interest:** The author declares no conflict of interest.

## Notes

[1] The Protestant American missionaries who arrived in the first five years of the twentieth century represented several denominations: Methodists, Episcopalians, Baptists, United Brethren, Disciples of Christ, Congregationalists, Christian and Missionary Alliance, and Seventh Day Adventists (Gowing 1967, p. 126).

[2] In the United States, it is common to refer to Protestantism as comprising two distinct categories: mainline denominations (such as Methodists, Episcopalians, and Presbyterians) and evangelical churches. The former are generally considered more progressive in theology and social issues than the more conversative evangelicals. These American divisions are also seen in the Philippines—there are similarly mainline denominational churches and evangelical congregations—but the distinction is less pronounced. In this article I use "Protestant" and "evangelical" more or less as synonyms, to approximate how the terms tend to be used in the Philippines. It should also be noted that roughly one-tenth of Filipinos are Muslim, a faith tradition that preceded by centuries the arrival of Spanish missionaries.

[3] The relationships of dominance among indigenous languages of the Philippines, as well as against the languages of the colonizers, is described by (Tupas 2015).

[4] For a history of American use of English in the educational system, including its use as a tool for colonization, see (Martin 2008).

[5] Kendall uses the concept of "syncretism" to describe the interplay between respective cultural materials of Spain and the Philippines (Kendall 2010). Irving frames those intercultural interactions as "counterpoint", with various participants working both together and against one another in the forging of a Hispanicized Filipino society (Irving 2010).

[6] Music localization is defined by Ingalls, Reigersberg, and Sherinian as "the process whereby Christian communities take a variety of musical practices—some considered 'indigenous,' some 'foreign,' some shared across spatial and cultural divides; some linked

7   to past practice, some innovative—and make them locally meaningful and useful in the construction of Christian beliefs, theology, practice, and identity" (Ingalls et al. 2018).

7   Cecila Herrera Marcelo was one such "adapter". Her son, Jungee Marcelo, notes that Cecila, who also worked on translations of scripture from Hebrew and Greek into Tagalog, never referred to her songs as translations. When it came to hymns, she called the task "adaptation".

8   My own translation of the Tagalog title would be something like, "Truly You Are Faithful".

9   A recording of this song, along with the rest that make up the first *Papuri* album, can be heard here: https://papurimusic.febc.ph/product/papuri-1/ (accessed on 30 August 2021).

10  Jungee Marcelo, a Filipino record producer who has worked with *Papuri*, stated that the 1980 song "Hesus" also mimics sonic features from 1970s artists such as Dan Fogelberg and Bread. He also notes that the song's interplay between major and minor chords is deliberately echoing similar chord progressions from Led Zeppelin's *Stairway to Heaven*. Interview on 4 May 2021.

11  Gerry and Sue Holloway were the Church of God (Cleveland, Tennessee) missionaries who planted the congregation in the early 1980s. https://wordfortheworld.com/about/history/ (accessed on 30 August 2021).

12  For differences in the various Pentecostal typologies of biblical worship architecture, see (Perez 2021).

13  Note that when considered a part of "praise and worship", the word "worship" takes on a specific meaning in contrast to "praise". Worship in this context is, therefore, not an overall term for one's response to God—it is rather a mode of singing that is often slower and more reflective of one's personal relationship with the Lord than the initial and upbeat "praise" section of songs.

14  For details about the International Worship Symposium, a series of conferences for praise and worship leaders in the 1970s and 1980s that taught about music as the primary medium of encounter with the divine, see (Perez 2021).

15  Lim and Ruth, *Lovin' on Jesus*, 113–14.

16  My own experience leading song-writing workshops throughout the Philippines in the first decade of the twenty-first century revealed that the "praise to worship" formula was widespread in urban and rural areas, across many denominational groups. In its simplest formulation, church musicians would plan three fast (*mabilis* in Tagalog) songs followed by three slow (*mabagal*) ones.

17  Interviews with Roce Anog, 29 September 2020, and Evelyn Martin, 14 October 2020.

18  This is not to say that Tagalog songs were never used at Word for the World. Some were used as "special numbers"—that is, performances by a soloist or ensemble that were observed, but not joined, by the congregation. These included compositions by *Papuri*. Word for the World also had specific services in Tagalog for people more comfortable using that language in worship, and those gatherings used Tagalog language congregational songs.

19  For a description of a megachurch that uses Tagalog in its services at a strategic attempt to appeal to working-class Filipinos, see (Cornelio 2018).

20  Take, for instance, Luhrmann's reduction of Pentecostalism to core components found in various nations: "The overt features of Pentecostalism—tongues, spiritual warfare, biblical literalism, and the direct immediacy of an encounter with God—make church practice clearly recognizable. In Accra and Chennai, these are colonial imports. They have been embraced with vigor" (Luhrmann 2020, p. 88).

21  https://www.kaloobdance.com/The_Founder.html (accessed on 14 May 2021).

22  http://www.redemptionofdance.org/The_Research.html (accessed on 14 May 2021).

23  http://www.redemptionofdance.org/About_Us.html (accessed on 14 May 2021).

24  There are several videos of this song available online. For one that includes of dancers performing to a recording of the song, see https://www.youtube.com/watch?v=idfLEW1nsSQ (accessed on 30 August 2021).

25  Translation by Ed Lapiz.

26  *Apo Dios* is the word for God in a wide variety of languages in the northern Philippines, including Ilocano, the trade language for northern Luzon. It is not a Tagalog word, but it would be widely recognized as a name for God in widespread use across the Philippines.

27  Written reports of early visitors to the islands note varied musical practices among the coastal cultural groups: "These reports cite various types of vocal genres, including epics relating genealogies and exploits of heroes and gods; work songs related to planting, harvesting, and fishing; ritual songs to drive away evil spirits, or to invoke blessings from good spirits; songs to celebrate festive occasions, particularly marriage, birth, victory at war, and the settling of tribal disputes; songs for mourning the dead; songs of courting; and children's game-playing songs. Musical instruments included those of bronze, wood, or bamboo—gongs, drums, flutes, zithers, lutes, clappers, and buzzers" (Canave-Dioquino 1998).

28  Nationalist composer José Maceda (b.1914) is one artist who embraced hybridization, composing new art pieces that quoted and referred to rhythms and melodies, even using traditional instruments alongside Western ones to create unique timbral settings. Some of these notable achievements were done with Christian worship in mind. One of Maceda's works, Pagsamba (Worship), set the standard text of the Roman Catholic Mass to an ensemble of two hundred musicians playing traditional instruments. Other composers, such as Francisco Feliciano (b.1941), also tapped into traditional music sounds, creating hybridized music for

liturgical settings and teaching generations of students to take up similar projects through the Asian Institute of Liturgy and Music (Santos 1998, pp. 875–76).

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
