# Peer review of "Protestant Congregational Song in the Philippines: Localization through Translation and Hybridization"

_religions, doi:10.3390/rel12090708_

Round 1

Reviewer 1 Report

Good writing here. It's an interesting selection of three different-but-related approaches to Christian music in the Philippines. I have some minor comments for revision, with the biggest question about the final section.

10: Word for the World, not Word for the Word

Footnote 2: Put separate quotation marks around “Protestant” and “evangelical”

93–97: Is that an overgeneralization at all? It seems like many colonial efforts included at least some people who were more innovative in engaging local cultural elements.

Footnote 7: I think put the final “adaptation” either in quotation marks or italics, to set it off as-a-term.

125–134: Are there other examples of Tagalog lyrics being mispronounced like this, besides just in translated hymns? Does it ever happen in pop songs?

136: “a serious movement”—were there any movements at all in that direction?

159: “strings”

161: “women’s voices”

Footnote 13: Colon after the single-quote

254: that-->than

311: “the work of the Holy Spirit could work”—awkward; remove one of the “work”s

402: Hyphen or comma in the song title? Consistency

313ff.: Do Filipinos understand performances by KALOOB as a new innovation, or a hearkening back to something from the past? I’m trying to understand what categories and concepts these performances occupy, and whether the reception differs in any way from the intentions of the performers. How does this performance context compare/contrast with other “cultural revival” genres in other parts of the world? I feel like there’s much more to say about this section of the paper—I’m seeing this as the most complex of the three case studies—but then it just ends.

Author Response

10: Word for the World, not Word for the Word: Corrected

Footnote 2: Put separate quotation marks around “Protestant” and “evangelical”: corrected

93–97: Is that an overgeneralization at all? It seems like many colonial efforts included at least some people who were more innovative in engaging local cultural elements: I think my original claim is generally true, but I made some modifications to make the language less totalizing.

Footnote 7: I think put the final “adaptation” either in quotation marks or italics, to set it off as-a-term.: Corrected

125–134: Are there other examples of Tagalog lyrics being mispronounced like this, besides just in translated hymns? Does it ever happen in pop songs? - I have never heard of this issue outside of translated hymns; Filipinos do not translate pop songs from other languages

136: “a serious movement”—were there any movements at all in that direction?: changed to "sustained" from "serious" to allow the possibility that their were some earlier attempts

159: “strings”: corrected

161: “women’s voices”: corrected

Footnote 13: Colon after the single-quote: corrected

254: that-->than: corrected

311: “the work of the Holy Spirit could work”—awkward; remove one of the “work”s : corrected

402: Hyphen or comma in the song title? Consistency: corrected

313ff.: Do Filipinos understand performances by KALOOB as a new innovation, or a hearkening back to something from the past? I’m trying to understand what categories and concepts these performances occupy, and whether the reception differs in any way from the intentions of the performers. How does this performance context compare/contrast with other “cultural revival” genres in other parts of the world? I feel like there’s much more to say about this section of the paper—I’m seeing this as the most complex of the three case studies—but then it just ends. -- added a paragraph to the end of this section to address these questions

Reviewer 2 Report

I suggest you correct the name of the Pentecostal congregation in the ,,Abstract” (Word for the World instead of Word for the Word)

Author Response

Corrected